# Tamoxifen Sensitizes Acute Lymphoblastic Leukemia Cells to Cannabidiol by Targeting Cyclophilin-D and Altering Mitochondrial Ca^2+^ Homeostasis

**DOI:** 10.3390/ijms22168688

**Published:** 2021-08-13

**Authors:** Miguel Olivas-Aguirre, Liliana Torres-López, Zeferino Gómez-Sandoval, Kathya Villatoro-Gómez, Igor Pottosin, Oxana Dobrovinskaya

**Affiliations:** 1Laboratory of Immunobiology and Ionic Transport Regulation, Centro Universitario de Investigaciones Biomédicas, Universidad de Colima, Av. 25 de Julio 965, Villa de San Sebastián, Colima 28045, Mexico; miguel.a.olivas@gmail.com (M.O.-A.); torres_liliana@ucol.mx (L.T.-L.); kathya8899@hotmail.com (K.V.-G.); 2Facultad de Ciencias Químicas, Universidad de Colima, Carretera Colima-Coquimatlán, km. 9, Coquimatlán 28400, Mexico; zgomez@ucol.mx

**Keywords:** acute lymphoblastic leukemia, cannabidiol, tamoxifen, mitochondria, calcium overload, mitochondrial permeability transition pore, cyclophilin D

## Abstract

Cytotoxic effects of cannabidiol (CBD) and tamoxifen (TAM) have been observed in several cancer types. We have recently shown that CBD primarily targets mitochondria, inducing a stable mitochondrial permeability transition pore (mPTP) and, consequently, the death of acute lymphoblastic leukemia (T-ALL) cells. Mitochondria have also been documented among cellular targets for the TAM action. In the present study we have demonstrated a synergistic cytotoxic effect of TAM and CBD against T-ALL cells. By measuring the mitochondrial membrane potential (ΔΨm), mitochondrial calcium ([Ca^2+^]_m_) and protein-ligand docking analysis we determined that TAM targets cyclophilin D (CypD) to inhibit mPTP formation. This results in a sustained [Ca^2+^]_m_ overload upon the consequent CBD administration. Thus, TAM acting on CypD sensitizes T-ALL to mitocans such as CBD by altering the mitochondrial Ca^2+^ homeostasis.

## 1. Introduction

Leukemia is the most commonly diagnosed cancer in children and adolescents. For 2021, there are more than 5690 new cases and 1600 deaths estimated for ALL in the United States [1]. T-lineage acute lymphoblastic leukemia (T-ALL) is the less common but highly aggressive ALL associated with unresponsiveness to chemotherapy, and poor prognosis with the lowest survival and highest recurrence rate when compared to other leukemic phenotypes [2,3]. In this context, multiple mechanisms of chemoresistance have been identified by which T-ALL survives after therapy, thus contributing to the frequent relapses and subsequent death of patients [4,5,6,7]. Despite the substantial progress made in antileukemic treatment [3,8,9], new strategies are needed to improve the therapy. There is accumulated evidence that a multitargeted therapy greatly improves the success rate as compared to monotherapies. 

The development of new drugs is a long-term and labor-consuming process, associated with substantial costs. Moreover, clinical trials often fall short of expectations due to a high toxicity and side effects of testing compounds. Therefore, drug repositioning represents an emerging attractive strategy [10]. Such an approach implies the search for novel molecular targets and mechanisms of already approved drugs, with the further expansion of their clinical use, which can significantly reduce the overall time and costs, with obvious benefits for the patients. CBD and TAM are FDA-approved drugs, already used to treat neurological disorders and estrogen receptor-positive (ER+) breast cancer, respectively [11,12,13]. However, it becomes progressively apparent that the clinical potential of these drugs can be extended. 

Recent evidence has demonstrated that CBD is a promising anticancer drug as it exhibits pronounced cytotoxicity against several cancer types, with a preference against T-ALL [14,15,16,17]. In addition to its anticancer properties, CBD has demonstrated broader benefits for cancer therapy through pain relief and enhancement of the cytotoxicity of anticancer drugs such as cisplatin, 5-fluorouracil or paclitaxel, a reduction of migration and metastasis and a limitation of the adverse effects of chemotherapy as in the case of doxorubicin use [18,19,20]. The antileukemic activity of TAM has been revealed in pre-clinical trials, using leukemic cell lines and primary cells, derived from leukemic patients [21,22,23,24,25]. TAM has also improved the leukemic response to all-trans retinoic acid [24], docetaxel [21], ceramide-centric therapies [26], romidepsin [27] and dexamethasone [25]. Leukemic cells are estrogen receptor (ER)-negative, suggesting alternative mechanisms of the TAM action. 

Unlike other cannabinoids, CBD does not act as an agonist of the cannabinoid CB1 or CB2 receptors. Alternative targets for CBD, including intracellular ones, have been recently identified [28]. In particular, we found that CBD modulates intracellular calcium ([Ca^2+^]) signaling and directly interacts with mitochondria, triggering leukemic cell death [16]. This cytotoxicity was due to the mitochondrial [Ca^2+^] overload, which leads to organelle dysfunction, mitophagy and cell death. 

Multiple “off targets” for TAM involve mitochondria. It stimulates the mitochondrial nitric oxide synthase activity and decreases oxygen consumption [29]. These effects were [Ca^2+^]_m_ dependent and a TAM-mediated [Ca^2+^]_m_ rise was observed in isolated mitochondria and, to a lesser extent and slower, in mitochondria within intact breast cancer cells. In addition, TAM promotes cell death by altering Ca^2+^ handling at different levels, including [Ca^2+^]_m_ [30,31]. Consequently, TAM is believed to alter the mitochondrial homeostasis [32].

There is accumulated evidence that mitochondria play a central role in leukemic progression. T-ALL cells have developed several mitochondrial adaptations, which enhance metabolism plasticity, avoid chemotherapy efficacy, and favor intracellular signaling. Therefore, drugs which target mitochondria (mitocans), are emergent tools for antileukemic treatments [33]. In the present work we addressed the effect of combined TAM and CBD application on the viability of T-ALL, with an emphasis on alterations in the mitochondrial function.

## 2. Results

### 2.1. CBD and TAM Act Synergistically to Decrease T-ALL Viability

The cytotoxicity of CBD and TAM against T-ALL was reported by our group earlier [16,25,34]. Here, to evaluate the possible synergism between these two drugs against leukemic cells, the cytotoxicity of CBD and TAM was evaluated either alone or in combination, on two leukemic cell lines (Figure 1a–d). Both cell lines exhibited a dose-dependent sensitivity to CBD and TAM. CBD cytotoxicity was more pronounced in CCFR-CEM as compared to Jurkat cells (Figure 1a,c). In contrast, TAM cytotoxicity was higher in Jurkat cells as compared to CCFR-CEM (Figure 1b,d). To test whether the drugs act synergistically, we pretreated the cells over 20 min with a fixed concentration of drug A, which provoked less than 20% of cytotoxicity and then tested the range of concentrations of drug B. When co-administered, TAM greatly enhanced the CBD cytotoxicity (Figure 1a,c; red traces). Likewise, CBD sensitized cells to TAM (Figure 1b,d; red traces). The effect was beyond the additive one. This is apparent when the experimental curves, which correspond to the joint drug applications, are compared with a prediction for a merely additive effect (Figure 1a–d, dotted lines). Moreover, the sequence of drug application matters. A more pronounced synergistic effect was observed, when TAM was applied first (Figure 1a,c). 

### 2.2. CBD and TAM Induce Cell Death

The observed decrease of the metabolic activity of the T-ALL cell population (measured by resazurin reduction) can be caused by a combination of the following factors: (1) inhibition of mitochondrial metabolism, (2) decreased proliferation rate or (3) increased cell death. Cell death, induced by CBD, TAM or CBD and TAM coadministration, was analyzed by confocal microscopy (Figure 1e–g). In these experiments, annexin V-conjugated with Alexa 488 (A488, green) was used as a marker of phosphatidylserine externalization during apoptosis, whereas propidium iodide (PI, red) was used to stain the cells with altered plasma membrane integrity. Then viable cells were double negative A488^−^PI^−^, while dead cells included the other three populations, namely A488^+^PI^−^ (early apoptotic), A488^−^PI^+^ (necrotic), or double positive A488^+^PI^+^ (late apoptotic and necrotic). The co-administration of CBD and TAM was more efficient in evoking cell death as compared to individual drug administration, and the synergistic effect increased at a longer (48–72 h) incubation. Interestingly, a combined action of CBD and TAM caused a marked increase of apoptosis and necrosis in Jurkat and CEM cells, respectively (Appendix A).

### 2.3. TAM Modifies the CBD Effect on Mitochondria

We previously reported that CBD interacts directly with mitochondria and causes mitochondrial Ca^2+^, [Ca^2+^]_m_, overload, stable opening of the mitochondrial permeability transition pore (mPTP) and consequent cell death [16,34]. TAM alters intracellular Ca^2+^ homoeostasis and likely interacts with mitochondria, among other targets [30,31,32,35]. Therefore, it was tempting to see whether TAM interfered with CBD at the mitochondrial level. 

To monitor [Ca^2+^]_m_, Jurkat cells were transfected with the mitochondria-targeted genetically encoded Ca^2+^-sensitive (*Kd* = 11 μM) indicator CEPIA3mt [36]. Transfection efficiency was confirmed by flow cytometry, and protein expression and localization were monitored by confocal microscopy. In successfully transfected cells, mitochondria were observed as multiple puncta (Figure 2a). Spectrofluorometric assay demonstrated that CBD rapidly evoked large [Ca^2+^]_m_ transients (Figure 2b, green trace). In cells, preincubated with the mPTP inhibitor cyclosporine A [37] (CsA, 10 μM, 20 min), the response to CBD was transformed from a transient to a sustained one (Figure 2b, purple trace). In the latter case_,_ based on the CEPIA3mt titration curve [36], [Ca^2+^]_m_ remained at a high (μM) level, indicating that mPTP opening was the mechanism responsible for [Ca^2+^]_m_ clearance after CBD-induced Ca^2+^ uptake. In contrast to CBD, TAM alone did not modify the [Ca^2+^]_m_ level (Figure 2b, red trace). However, in cells preincubated with TAM (7.5 μM, 20 min), CBD evoked [Ca^2+^]_m_ response, which was significantly higher in amplitude and stable in time (Figure 2b, black trace) and reminiscent, but in excess of that evoked by a combination of CBD and CsA. This result suggests that TAM prevents the mPTP formation. The same type of [Ca^2+^]_m_ response was observed when CBD and TAM were added simultaneously (Figure 2b, blue trace), indicating an instant TAM interference with the CBD-induced [Ca^2+^]_m_ response. When cells were preincubated with CBD (30 μM, 20 min), e.g., when the CBD-induced [Ca^2+^]_m_ transient relaxed and [Ca^2+^]_m_ returned to its resting level, neither TAM nor CsA produced any significant change in [Ca^2+^]_m_ (Figure 2b). Peak [Ca^2+^]_m_ value and steady state level at 500 s recorded for each condition are shown in Figure 2c. Collectively, these data support the view that TAM prevents the mPTP formation, so that a posterior CBD-induced Ca^2+^ uptake by mitochondria results in a stable enhanced [Ca^2+^]_m_.

[Ca^2+^]_m_ overload is the primary mechanism, leading to mPTP formation. Once the mPTP is formed and remains stably open, the mitochondrial membrane potential (ΔΨm) collapses. [37,38]. To monitor ΔΨm, Jurkat cells were stained with tetramethylrhodamine ethyl ester, TMRE (200 nM; Ex: 488 nm; Em: 575 nm), a cationic dye, which is retained in energized mitochondria. Simultaneous staining with MtGreen (Ex: 488 nm; Em: 510 nm) demonstrated a co-localization of the two dyes in mitochondria (Figure 2d). 

CBD administration caused a rapid decrease of the TMRE fluorescence intensity, indicating the collapse of the ΔΨm (Figure 2e; green trace). A preincubation with CsA (10 μM, 20 min) limited the CBD-induced ΔΨm loss (Figure 2e, purple trace). When administered alone, TAM did not have any significant effect on the ΔΨm during the 500 s recorded (Figure 2e, red trace). However, TAM significantly reduced the ΔΨm loss, induced by CBD, regardless of whether it was added before or simultaneously with CBD (Figure 2e, blue and black traces). The ΔΨm loss, induced by CBD (30 μM, 20 min), was partly reversed by posterior application of CsA or TAM (Figure 2e; yellow and pink traces). Mean TMRE fluorescence before and after (steady state at 500 s) application of a single or second drug are plotted in Figure 2e,f, respectively. Thus, ΔΨm depolarization was due to mPTP formation and can at least partly be reversed by the mPTP inhibition by CsA, whereas the eventually achieved steady-state of [Ca^2+^]_m_ depended on the sequence of the drug application. If the lock of the mPTP occurred after the relaxation of the CBD-induced [Ca^2+^]_m_ rise, [Ca^2+^]_m_ stayed at its resting level. If the mPTP formation was prevented prior to the CBD treatment, mitochondrial Ca^2+^ uptake evoked by the latter resulted in a stable high [Ca^2+^]_m_.

### 2.4. TAM Limits the mPTP-Mediated Cyt-c Release

The mPTP opening induces a rapid ΔΨm loss, uncoupling of cell metabolism, depletion of ATP and mitochondrial cristae remodeling, mitochondrial swelling and permeabilization, and Cyt-c translocation to the cytosol [39,40]. To explore the effects of TAM on CBD-induced Cyt-c release, Jurkat cells were transfected with a fluorescent tagged Cyt-c [41] and stained with MtRed to confirm the mitochondrial Cyt-c localization. Untreated Jurkat cells were characterized by a perfect co-localization of EYFP and MtRed (Figure 2g). CBD (30 μM, 1 h) rapidly induced the Cyt-c translocation to the cytosol, observed as a diffuse staining within the cell. As expected, the mPTP inhibitor CsA (10 μM) limited the CBD-induced Cyt-c release. The effect of CsA was mirrored by that of TAM (7.5 μM, 20 min), suggesting a similar mechanism of the TAM and CsA action on mPTP formation. 

### 2.5. TAM Can Interact with CypD to Inhibit the mPTP: In Silico Evidence 

The mPTP formation is driven by the recruitment of several proteins associated with the inner and outer mitochondrial membranes. Even though the protein composition of the mPTP remains controversial, there is a consensus that CypD is an obligatory component [37,40,42,43,44]. CsA, which directly interacts with CypD, is a potent and universal inhibitor of mPTP formation. 

Different cellular cyclophilin isoforms possess the CsA-binding domain (CsABD), which was characterized in detail by means of site-directed mutagenesis (reviewed in [43]). Basing on this information, several nonpeptidic CypD inhibitors with a different affinity and selectivity, targeting CsABD, were developed, ([45,46]; Figure 3b). To test whether CsABD is a potential TAM-binding site, protein-ligand interaction analysis was performed. For a comparison, docking within the human CypD (PDB: 2ZEW) for TAM and two TAM metabolites, and six nonpeptidic CypD-inhibitors (Fragment 3, 4, 7, 8, 14, and 40) with variable affinity and defined interaction sites [45] were tested in silico (Figure 3). All the aforementioned molecules tended to interact with two particular regions, defined as S1′ and S2′ pockets, which correspond to the CypD catalytic site (Figure 3a, green areas). The predicted coordination of nonpeptidic inhibitors by certain amino acid residues was in a good agreement with previously published results [45,47]. The table in Figure 3b reveals a correlation between experimentally defined binding affinity and binding energy for these compounds. It may be presumed, then, that TAM and its metabolites, which are characterized by an even higher binding energy, may also have a higher affinity to the CsABD, perhaps, comparable to that of the CsA (*Kd* = 30 nM). A higher predicted binding energy for TAM and its metabolites is due to the fact that these can bind not only to Gln 63 and Phe 113, conserved among all the evaluated molecules, but also to Phe 60, Ala 101, Asn 102, Leu 122 and His 126, resulting in a larger number of interacting residues. Therefore, the TAM-mediated mPTP inhibition may be due to the direct CypD-TAM interaction at a specific CsA-binding site. 

## 3. Discussion

Mitochondria are central regulators of cancer cell viability and progression. In this study, we demonstrated by different approaches, that CBD and TAM target mitochondria to promote cell death by convergent pathways. CBD cytotoxicity relies on its capacity to produce mitochondrial Ca^2+^ overload, mPTP opening, Cyt-c release, decrease of metabolic activity, and cell death via apoptosis and mPTP-driven necrosis (Figure 1; Figure 2 of this study; [16]). TAM has been demonstrated to possess cytotoxic effects on several cancer types, including T-ALL [21,22,23,25]. However, for T-ALL the precise mechanism by which TAM promotes cell death is not completely understood. Traditionally, TAM effects are attributed to the modulation of the estrogen receptors, intracellular ERs: α/β or plasmatic membrane GPER. T-ALL cells only express GPER [25]. At the same time, a specific GPER antagonist G-36 did not prevent TAM-cytotoxicity, which suggests additional cellular targets. The present study on T-ALL cells demonstrates that TAM is rapidly (within seconds) incorporated into mitochondria and prevents mPTP formation, induced by CBD (Figure 2). Our data agree with the results by other groups, which show that TAM targets isolated rat mitochondria and impeded mPTP formation, thus mimicking the effect of CsA [30]. The main target for CsA are cyclophilins. CypD, the cyclophilin isoform, expressed in mitochondria, is an obligatory component of the mPTP. Thus, CsA binding to CypD inhibits the mPTP [37]. The fact that CsA and TAM effects on [Ca^2+^]_m_ and ΔΨm are fully comparable, urged us to perform a comparative docking analysis for binding of TAM and its metabolites, CsA and several nonpeptidic CypD inhibitors. Our analysis showed that all these compounds bind to the CypD active site and that the predicted strength of the TAM interaction exceeds that for the nonpeptidic inhibitors and, likely, approaches that for CsA (Figure 3). The proposed interacting amino acid residues for nonpeptidic CypD inhibitors coincided with those pinpointed by others [45,46,47]. All tested compounds, including TAM, share Gln 63 and Phe 113, but TAM can interact with additional residues, which tends to increase its binding affinity. Therefore, like CsA, TAM can prevent the mPTP formation by the arrest of CypD integration. 

The synergistic effect of CBD and TAM surprisingly depended on the sequence of drug application, being greater in the case of CBD after TAM (Figure 1a–d). The clues for this difference may be found in Figure 2. When it came to ΔΨ_m_ the result was the same, i.e., a steady state depolarization, intermediate between that caused by CBD and TAM alone, independent of the sequence of their application. Contrary to this, if the mPTP opening by TAM was impeded before or after CBD treatment, [Ca^2+^]_m_ was fixed at rather high (micromolar) or resting levels, respectively. A stable [Ca^2+^]_m_ overload of this magnitude is a poorly explored state, because normally upon such an increase the Ca^2+^ is rapidly cleaned via the mPTP, here prevented by CsA or TAM. Prevention of the mPTP formation by cytotoxic CsA concentration after long (>8 h) exposure alone causes an increase of both cytosolic and mitochondrial Ca^2+^, from 0.1 to 0.5 μM and ΔΨ_m_ collapse, which results in the ATP depletion [48]. Similarly, TAM at long (48 h) incubation provoked a moderate [Ca^2+^]_m_ increase in intact breast cancer cells, albeit that it caused faster and greater [Ca^2+^]_m_ responses with isolated mitochondria [29]. One may expect a higher cytotoxic effect for TAM/CBD co-administration, when even higher [Ca^2+^]_m_ is reached almost instantaneously. The sustained Ca^2+^ overload should inevitably affect mitochondrial metabolism and ATP synthesis. Considering the effects of TAM on mPTP formation, one important question is which cell death scenario is established upon CBD/TAM co-treatment. As mPTP cannot be formed, alternative mechanisms leading to cell death can be invoked such as outer membrane permeabilization (MOMP), mediated mainly by pro-apoptotic members of the BCL-2 family and, to a lesser extent, the autophagy [49], to be addressed in future research. 

CypD activity is not limited to mPTP formation. CypD plays important roles in protein folding, as chaperone, regulating the OXPHOS activity, among others [50]. It is not surprising then, to find several independent reports of CsA/TAM effects as great autophagy inductors. Such observations include the upregulation of beclin 1, LC3 II, and the presence of multiple autophagosomes in different cell lines [25,49,51,52]. On the other hand, CsA affects all cellular cyclophilins, including cytosolic CypA. The well-known immunosuppressive effect of the CsA was explained via the CypA-CsA complex, inhibiting the calcineurin A and NFAT-regulated pathway [53,54]. Our data suggest that the molecular targets for TAM and CsA may overlap more than has been expected. Thus, it should be tested on human cell models whether TAM can affect the calcineurin. Of note, the antifungal action of TAM was explained by the inhibition of the calmodulin binding to calcineurin [55,56]. Calmodulin is Ca^2+^-binding protein, which carries four canonical Ca^2+^-binding sites (EF-hands) and TAM was shown to interact directly with calmodulin EF-hands [57]. 

The main findings of this work are summarized in Figure 4. We have demonstrated that TAM most likely interferes with the CBD action by a prevention of mPTP formation, similar to the CsA effect. In silico analysis shows that TAM and CsA share the same binding site within the catalytic center of the CypD. Depending on the sequence of drugs, TAM and CBD application, two distinct mitochondrial states are generated, with high (TAM first) or resting (CBD first) [Ca^2+^]_m_. Both states are characterized by the same level of ΔΨm depolarization. In both cases, TAM and CBD displayed a synergistic action against T-ALL, which was higher in the case of the TAM first, CBD second application sequence, where a high [Ca^2+^]_m_ state was generated. Collectively, our results suggest that a combination of TAM and CBD offers an attractive strategy to improve the T-ALL therapy. 

## 4. Materials and Methods 

### 4.1. Reagents

CBD (Cayman Chemicals; Ann Arbor, MI, USA; Cat. #90081) and TAM (Sigma-Aldrich; San Luis, MO, USA; Cat. #T5648), were employed in this study. Stock solutions were stored at −20 °C before use. The used solvent, ethanol or methanol, concentrations did not affect cell viability. 

### 4.2. Cell Lines and Culture Conditions 

Leukemic cells from T-cell acute lymphoblastic leukemia CCFR-CEM (CCL-119) and Jurkat (Clone E61, TIB-152) were obtained from the American Type Culture Collection (ATCC; Manassas, VA, USA). Cells were cultured in RPMI 1640 medium, supplemented with 10% of heat-inactivated fetal bovine serum (FBS), 2 mM Glutamax, 10 mM HEPES, 100 U/mL penicillin, and 100 μg/mL streptomycin) (all from Gibco, Thermo Fisher Scientific, FairPoint, NY, USA). Cell cultures were maintained in a humidified incubator (37 °C, 5% CO_2_). Culturing was restricted to the first 20 passages. 

### 4.3. Viability Assay

Cells were collected, centrifuged, and resuspended in fresh media to a final concentration of 1 × 10^6^ cells/mL. CBD or TAM were added to cell cultures for 24 h. For synergism experiments, the drug A at fixed concentration was added 20 min prior to addition of variable concentrations of the drug B. After this, 180 μL of cells and 20 μL of the Tox 8 reagent (Sigma-Aldrich) were mixed and incubated for 4 h. This assay is based on the fact that nonfluorescent reagent (resazurin) is intracellularly reduced into a highly fluorescent molecule (resorufin) by metabolically active cells. Resorufin fluorescence was further estimated by a GloMax plate reader (Promega, Madison, WI, USA) by exciting each sample at 525 nm and collecting the fluorescence at 580–640 nm. Results from independent experiments were averaged and normalized to the control group. 

### 4.4. Cell Death Analysis 

Dead cell apoptosis kit from Thermo Fisher Scientific (V13241) was used as recommended by manufacturer, with some modifications. The kit contains a marker for necrosis (propidium iodide; PI, ex. 535 nm, em. 617 nm) which is nucleophilic and only stains the cells with plasma membrane damage. The kit also contains a marker for apoptosis (Annexin V-Alexa Fluorv488), which stains phosphatidylserine, a phospholipid from the inner layer of the plasmatic membrane that is externalized upon the apoptotic induction by deregulation of the flippases and scramblases activity. For the experiments, 1 × 10^6^/mL cells were treated with the determined concentrations of CBD, TAM, or their combination for 24, 48 or 72 h. After the treatment, cells were collected, drug and RPMI were removed by centrifugation (100× *g*, 5 min) and cell pellet was resuspended in PBS. For each 1 × 10^6^ cells, 3 μL of Annexin and 1 μL of PI (working solution 200 μg /mL) were added and incubated for 20 min dissolved in 100 μL of 1X Annexin binding buffer. Specific fluorescence was evaluated by means of an LSM 700 confocal microscope (Carl Zeiss, Jena, Germany). Acquired images were further analyzed in ImageJ software (NIH, download available online) by estimating the percentage of apoptotic, necrotic, or double positive cells for a randomly selected field. At least 50 cells per field were evaluated and the data represent the average of 5 independent experiments for each condition. 

### 4.5. Leukemic Cell Transfection with CEPIA3mt or EYFP-Cyt-c

CEPIA3mt/pCMV (36) or EYFP-Cyt-c (41) construct was added to competent bacteria (DH5a; Thermo Fisher Scientific, Cat. 18258012) and were further transformed by heat shock. Bacterial culture was incubated for 14 h at 37 °C in LB agar (Thermo Fisher Scientific, Cat. 22700025) and selected by ampicillin administration (100 μg/mL; Gibco, Thermo Fisher Scientific, FairPoint, NY, USA Cat. 11593027). Selected colonies were cultured for 14h to promote bacterial growth. Next, the NucleoBond XtraMidi (Machery-Nagel, Düren, Germany, Cat. 740410.10) was employed to obtain the purified plasmidic DNA. Quantitation of the obtained DNA material was estimated by spectrophotometry (reading the absorption at 260/280 nm). A sample of 10^5^/mL of Jurkat cells were cultured under reduced OptiMem medium for 12 h to promote starvation and a consequent increased plasmid uptake (Master mix composed of Lipofectamine 3000 and 1 μg of plasmidic DNA for CEPIA3mt and 500 ng of plasmidic DNA in the case of EYFP-Cytc-c). Cells were centrifugated (400× *g*, 30 min) to promote the interaction of loaded liposomes and cells. Then, cells were incubated overnight and 10% of FBS was added in the next day. CEPIA3mt expression was evaluated by flow cytometry (FACS Canto II, BD Biosciences, Franklin Lakes, NJ, USA) and confocal microscopy to determine the transfection efficiency. Near to 98% of the collected events were positive to CEPIA3mt (ex. 488 nm, em. 510) by flow cytometry. Selective mitochondrial localization of CEPIA3mt was confirmed by confocal microscopy (LSM700, Carl Zeiss) using a Z-stack analysis of transfected cells to observe the distribution and co-localization with mitochondrial dyes as TMRE and Mitotracker Red FM. EYFP-Cyt-c expression was evaluated by flow cytometry (FACS Canto II, BD Biosciences) and confocal microscopy (LSM 700, Zeiss, Jena, Germany Ex. 514 nm, Em. 525 nm) to determine the transfection efficiency. 

### 4.6. Mitochondrial Ca^2+^ Measurements 

CEPIA3mt-transfected Jurkat cells were employed 24 h after transfection. A sample of 1 × 10^6^/mL of leukemic cells were collected and resuspended in Hanks’ balanced salt solution (HBSS; NaCl 143 mM, KCl 6 mM, MgSO_4_ 5 mM, HEPES 20 mM, BSA 0.1%, glucose 5 mM, EGTA 1 mM, pH 7.4, ≈300 mOsm) and added into a quartz cuvette. Samples were evaluated in a HITACHI F7000 spectrofluorometer (Hitachi High Tech, Tokio, Japan) by exciting at 488 nm and collecting CEPIA3mt fluorescence at 510 nm every 2.5 s. Traces from independent experiments were normalized to initial CEPIA3mt fluorescence (F/F0) and averaged.

### 4.7. Evaluation of the Mitochondrial Membrane Potential

Jurkat cells (1 × 10^6^ /mL) were collected, washed, resuspended in HBSS, anCEPIA3mtd stained with the mitochondrial membrane potential indicator TMRE (Thermo Fisher Scientific, Cat. T669; 200 nM, 30 min). Then, cells were washed with HBSS again to eliminate the extracellular TMRE. Selective TMRE staining was confirmed by confocal microscopy (LSM 700) by co-staining with the mitochondrial tracer MtGreen (Thermo Fisher Scientific, Cat. M7514; 100 nM, ex. 490 nm, em. 518 nm). To monitor the ΔΨm in time, stained cells were transferred to a quartz cuvette and TMRE fluorescence was recorded every 2.5 s before and up to 10 min after the drug application, using HITACHI F7000 spectrofluorometer (Ex. 555 nm, em. 582 nm). In case of sequential drug application, the first drug was added 20 min before the second one. Data are mean TMRE fluorescence intensity for 3 independent experiments in each condition.

### 4.8. In Silico Protein-Ligand Interaction 

Potential TAM interaction with human CypD was analyzed in silico with the use of Molegro Virtual Docker 6.0 software (Molexus IVS, Odder, Denmark). Original paper, describing molecular docking algorithm, employed by MVD, can be found in: Thomsen and Christensen, 2006. Chemical structures of CsA (2909), TAM (2733526), 4-hydrotamoxifen (449459) and endoxifen (10090750) were obtained from PubChem Database (NIH; https://pubchem.ncbi.nlm.nih.gov, accessed on 25 January 2021) and the structure of human CypD (2Z6W) and its inhibitors (6R9S, 6R9U, 6RA1, 6R9X, 6R8O, 6R8W) were acquired from the Protein Data Bank (PDB; https://www.rcsb.org/; accessed on 25 January 2021; [58]). Molecules were loaded and independent dockings for each ligand were performed against 2Z6W. First, the cavities corresponding for S1′ and S2′ pockets were identified, and a custom search space was defined to improve the docking accuracy. MolDock Score was selected as a scoring function and internal electrostatic interactions (ES) and hydrogen bond interactions (HB) were addressed. MolDock Optimizer was selected as a search algorithm and 20 runs (number of times that the docking simulation is repeated for each ligand) were chosen. Iteration and population size were set as recommended by the *Docking wizard* tool and the best 5 poses were requested and analyzed. The docking was validated by comparing the affinities and predicted binding sites for selected ligands with the CypD amino acid residues from an independent study (Grädler et al., 2019). Contribution of individual CypD residues to ligand binding were evaluated by the ligand energy inspector (*L.E. inspector/Targets*). According to their binding energy, interactions can be divided into main (<−20 units), strong (−20 to −10), intermediate (−10 to −5), and weak (−5 to 0).

### 4.9. Cyt-c Release Evaluation

EYP-Cyt-c transfected cells were evaluated by confocal microscopy (LSM 700; ex. 514 nm, em. 526 nm). The 10^5^ Jurkat cells were treated for 1 h by CBD, TAM or their combination. After this, cells were washed and resuspended in PBS, placed in a homemade record chamber, and the images were acquired from selected fields, using a 63× oil-immersion objective. To confirm the Cyt-c localization inside or outside the mitochondria, Mitotracker Red FM was used.

## Figures and Tables

**Figure 1 ijms-22-08688-f001:**
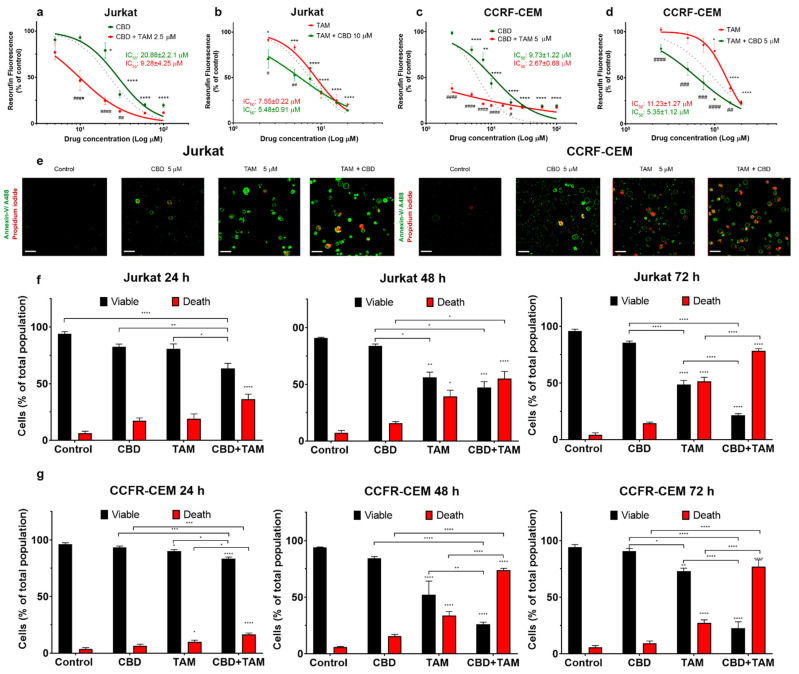
Synergistic cytotoxicity of CBD and TAM against T-ALL cell lines. The cytotoxic effect of CBD (0–100 μM) and TAM (0–20 μM) was estimated in Jurkat (**a**,**b**) and CCFR-CEM cells (**c**,**d**) at 24 h by measuring resorufin production. Data points are mean ± S.E of at least 3 independent experiments normalized to control group. One-way ANOVA/Dunnett’s multiple comparison test was used for statistical analysis for the dose response curve of single drugs compared to control group (* *p* < 0.05; ** *p* < 0.01; *** *p* < 0.001; **** *p* < 0.0001). Two-way ANOVA/Bonferroni’s multiple comparison test was used for comparison of the corresponding concentrations between single and co-administration (# *p* < 0.05; ## *p* < 0.01; ### *p* < 0.001; #### *p* < 0.0001). Dotted lines are predictions under assumption that drugs act independently and additively. (**e**–**g**) Confocal microscopy analysis of cell death induced by CBD (5 μM), TAM (5 μM), or their combination. After treatment, cells were stained with annexin V conjugated to Alexa488 (A488; Ex: 488 nm, Em: 510 nm) and propidium iodide (PI; Ex: 535 nm, Em: 617 nm). (**e**) Representative micrographs of Jurkat (**left**) or CCFR-CEM (**right**) cells treated for 24 h. For statistical analysis, the data were collected at every time point (24, 48 and 72 h) and present as viable (A488^−^PI^−^), or dead cells, which include apoptotic (A488^+^PI^−^), necrotic (A488^−^PI^+^) or double positive (DP, A488^+^PI^+^) cells in Jurkat. Scale bar represents 20 μm. (**f**) and CCRF-CEM (**g**) cell populations. Data are mean ± S.E. of at least 3 independent experiments (summarizing at least 180 cells analyzed for each condition). Two-way ANOVA and multiple comparison test Tukey was employed to determine statistical differences between groups (color coded to identify compared groups; * *p* < 0.05; ** *p* < 0.01; *** *p* < 0.001; **** *p* < 0.0001).

**Figure 2 ijms-22-08688-f002:**
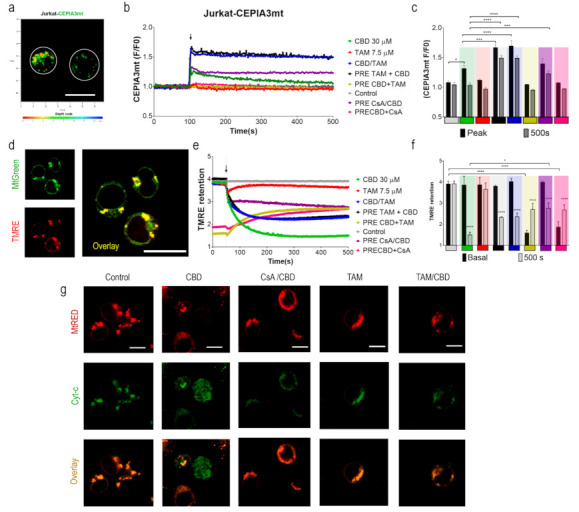
Effects of TAM and CBD on [Ca^2+^]_m_ and ΔΨ_m_ in Jurkat cells. (**a**) Representative image of CEPIA3mt-transfected cells, colored puncta are mitochondria. Scale bar is equivalent to 10 μm. (**b**) Time course of [Ca^2+^]_m_ changes upon CBD or TAM administration. (**c**) Quantification of the peak value and the [Ca^2+^]_m_ level at 500 s obtained from transients upon drug administration. Data from every graph represent the average of at least 4 independent experiments. Two-way ANOVA and Sidak’s multiple comparison test were employed to determine statistical differences between groups (color coded to identify compared groups; * *p* < 0.05; ** *p* < 0.01; *** *p* < 0.001; **** *p* < 0.0001). (**d**) Representative images of TMRE and MtGreen co-stained Jurkat cells evaluated by confocal microscopy, scale bar represents 10 μm. (**e**) Monitoring of ΔΨm upon CBD, TAM or their combinations. Pretreatment was given 20 min before second drug administration. (**f**) Quantification of the basal TMRE retention and TMRE levels upon treatments (initial and at 500 s). Data represent the average of at least 4 independent experiments. Two-way ANOVA and Sidak’s multiple comparison test were employed to determine statistical differences between groups (* *p* < 0.05; ** *p* < 0.01; *** *p* < 0.001; **** *p* < 0.0001). (**g**) Representative images of Jurkat-EYFP-Cyt-c cells treated with TAM (7.5 μM), CBD (30 μM), or TAM + CBD treated cells by confocal microscopy (1 h). Cyt-c distribution corresponds to MtRed staining as multiple discrete puncta. Scale bar corresponds to 10 μm.

**Figure 3 ijms-22-08688-f003:**
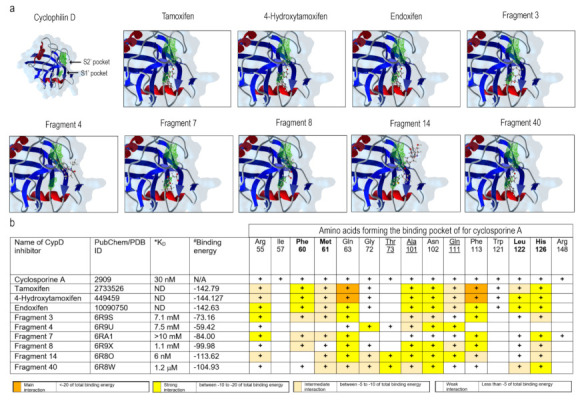
In silico evaluation of the molecular interactions of human CypD with TAM, TAM derivatives or CypD nonpeptidic inhibitors. (**a**) Structure of CypD (Left; PDB: 2Z6W), green shadowed regions represent the S1′ and S2′ pockets of human CypD. Predicted interactions and orientation of selected ligands within CypD are drawn. (**b**) Summary of the obtained interaction energy values (MolDock Score), in vitro dissociation constant values (from [45]) and CypD residues’ binding contributions for different ligands. (+) implies that amino acid is involved in binding. Different strengths of interaction are color coded. * determined from surface plasmon resonance (SPR) and protein-based NMR studies. # Energy from interactions predicted by the docking analysis (Epair from MolDockScoring). **Bold** residues are part of the CypD active site and S1′ pocket. Underlined residues are part of the active site of CypD and the S2′ pocket.

**Figure 4 ijms-22-08688-f004:**
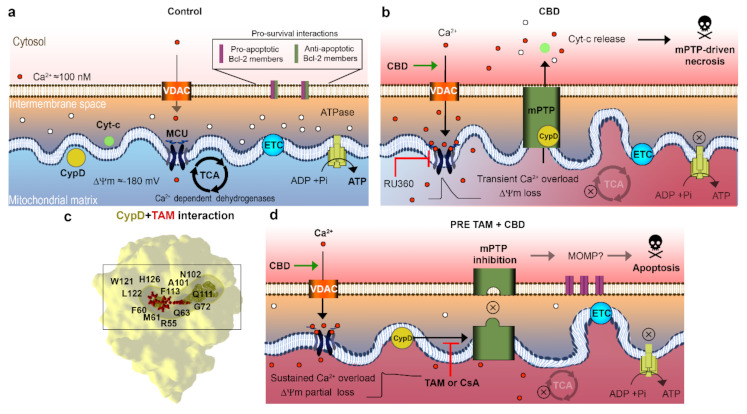
Summary of TAM and CBD effects on T-ALL mitochondria and cell fate. (**a**) Normally functioning mitochondria. A controlled Ca^2+^ entry stimulates the operation of tricarbonic acid cycle (TCA), hence the electron transfer by ETC, mitochondrial energization (high ΔΨm) and ATP synthesis by F-ATPase. Retention of pro-apoptotic factors like Cyt-c and a balance between pro- and anti-apoptotic members of the Bcl-2 protein family in the outer membrane. (**b**) CBD favors the Ca^2+^-highly permeable state of VDAC, promotes a transient [Ca^2+^]_m_ overload via MCU, stable formation of the mPTP, ΔΨm and energy collapse, release of Cyt-c, apoptosis and necrosis. (**c**) Predicted molecular interactions of TAM with CypD (PDB: 2Z6W) indicated amino acid residues coordinate both TAM and CsA binding. (**d**) A pretreatment with TAM prevents mPTP formation by arrest of the CypD integration and reduces but does not abolish the CBD-induced ΔΨm depolarization. It causes a sustained [Ca^2+^]_m_ overload upon CBD application, thus altering mitochondrial Ca^2+^ homeostasis, metabolism and ATP synthesis. A synergism between TAM and CBD favors apoptosis, which, in the absence of mPTP is likely mediated by a permeabilization of the outer membrane (MOMP) and respective release of pro-apoptotic factors. Dark arrows are for a direction of a transport or signaling process. When a black arrow is used instead of a grey one for the same process it implies an up-regulation. Green arrows imply that the process is stimulated (by CBD). Red lines with a bar head imply inhibition (by TAM or Ru360).

## Data Availability

The data presented in this study are available within the article and Appendix A.

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
