# Peer review of "Tamoxifen Sensitizes Acute Lymphoblastic Leukemia Cells to Cannabidiol by Targeting Cyclophilin-D and Altering Mitochondrial Ca2+ Homeostasis"

_ijms, 2021, doi:10.3390/ijms22168688_

Round 1
Reviewer 1 Report
I have reviewed the article by Olivas-Aguirre and colleagues “Tamoxifen Sensitizes Acute Lymphoblastic Leukemia Cells to 2 Cannabidiol by Targeting Cyclophilin-D and Altering Mito- 3 chondrial Ca2+ Homeostasis”. In this article, the authors studied the anti-leukemic effect of CBD and TAM in T-ALL leukemia cell lines. They showed that the co-administration of both drugs results in a synergic anti-leukemic effect involving mitochondria and Ca+2 levels. This synergy maybe due to the interaction of TAM with CypD, which avoids the formation of the mitochondrial pore to release Ca+2. The design and the methodology of the study are clear and answer the research question. The findings are interesting, which lead the authors to propose a mechanism of action of these drugs. On the other hand, there are several points that should be clarify a bit to facilitate the reading and understanding.
Major comments
Results
- Section 2.2 CBD and TAM induce cell death. The authors barely explain the results, instead they mention the methodology. Explanation about the methods help the reader to understand the experiment, but a more extended description of the results is missing, especially considering that the authors extract important conclusions to discuss in the discussion section (for instance, the second paragraph of the discussion starts with “Synergistic effect of CBD and TAM surprisingly depended on the sequence of drug application, being larger in case CBD after TAM (Figure 1 a-d)”, a result not mention in the results section)
- According to the results in section 2.3, TAM alone does not modify Ca+2 levels
Discussion
- As mentioned before, second paragraph of this section describes results that should be included in the previous section (They cite a supplementary figure with results not described before). In general, the discussion is not the place to show additional results.
- According to the results in section 2.3, TAM alone does not modify Ca+2 However, the authors state in the introduction that “the mechanism of TAM-mediated [Ca2+]m rise was not elucidated. In addition, TAM promotes cell death by altering Ca2+ handling at different levels, 69 including [Ca2+]”. It would be interesting to mention this in the discussion and provide possible explanations (maybe in a “Limitations” section at the end of the manuscript?)
- Figure 4c seems more appropriate in the results section.
- The legend in figure 4d is a very important part of the manuscript resuming the mechanism proposed by the authors and it should be mention along the discussion (maybe as a conclusion?)
Methods
- Sections 4.5 and 4.9 should be fused, since they described the transfection of different constructs and seems repetitive.
Author Response
Reviewer 1
I have reviewed the article by Olivas-Aguirre and colleagues “Tamoxifen Sensitizes Acute Lymphoblastic Leukemia Cells to 2 Cannabidiol by Targeting Cyclophilin-D and Altering Mito- 3 chondrial Ca2+ Homeostasis”. In this article, the authors studied the anti-leukemic effect of CBD and TAM in T-ALL leukemia cell lines. They showed that the co-administration of both drugs results in a synergic anti-leukemic effect involving mitochondria and Ca+2 levels. This synergy maybe due to the interaction of TAM with CypD, which avoids the formation of the mitochondrial pore to release Ca+2. The design and the methodology of the study are clear and answer the research question. The findings are interesting, which lead the authors to propose a mechanism of action of these drugs. On the other hand, there are several points that should be clarify a bit to facilitate the reading and understanding.
Many thanks for this reviewer for calling our attention to these specific points. According changes were made in the text.
Major comments
Results
- Section 2.2 CBD and TAM induce cell death. The authors barely explain the results, instead they mention the methodology. Explanation about the methods help the reader to understand the experiment, but a more extended description of the results is missing, especially considering that the authors extract important conclusions to discuss in the discussion section (for instance, the second paragraph of the discussion starts with “Synergistic effect of CBD and TAM surprisingly depended on the sequence of drug application, being larger in case CBD after TAM (Figure 1 a-d)”, a result not mention in the results section)
Accepted/explained. First, the effect of sequential application, seen in Figure 1 a-d corresponds to the 2.1 section. For the rest of Figure 1 the drugs, when combined, were applied simultaneously. We agree that the effect of the sequence of TAM and CBD application has be commented already in results, so we add two sentences at the end of section 2.1 (lines 95-97).
- According to the results in section 2.3, TAM alone does not modify Ca+2levels
Explained. Not at these times, perhaps not for this model and not in the range of micromoles (Kd for CEPIA3mt is 11 μM). We cannot exclude, however, slower and less pronounced increase of mitochondrial Ca2+ to be induced by TAM as observed in other cancer cellular models (see below).
Discussion
- As mentioned before, second paragraph of this section describes results that should be included in the previous section (They cite a supplementary figure with results not described before). In general, the discussion is not the place to show additional results.
Accepted. We removed this sentence from the Discussion and briefly commented the supplementary figure at the end of chapter 2.2 in the Results section.
- According to the results in section 2.3, TAM alone does not modify Ca+2 However, the authors state in the introduction that “the mechanism of TAM-mediated [Ca2+]m rise was not elucidated. In addition, TAM promotes cell death by altering Ca2+ handling at different levels, 69 including [Ca2+]”. It would be interesting to mention this in the discussion and provide possible explanations (maybe in a “Limitations” section at the end of the manuscript?)
Accepted/explained. First, we have mentioned in the introduction that TAM induced a rapid increase of free Ca2+ concentration in isolated mitochondria, being less pronounced and slower in mitochondria in intact breast cancer cells (lines 68-69). Now we mentioned it again also in the Discussion, just after discussing the effect of CsA on mitochondrial and cytosolic (lines 292-294). Using this analogy, we cannot exclude moderate and relatively slow Ca2+ changes, hypothetically induced by TAM in our cell model. These changes maybe barely below a detection limit by CEPIA3mt. We do not think that this is a limitation, because in this paper we have focused primarily on direct mechanisms of CBD and TAM action on mitochondria, which are involved in early response.
- Figure 4c seems more appropriate in the results section.
We considered this point. But 4c section in fact summarizes the interacting residues, already tabulated in Fig. 3b of results. So, it will be redundant in the Results section, but in our opinion appropriate in Fig. 4 and graphical abstract, just illustrating the binding site in a simpler way, easy to grasp visually.
- The legend in figure 4d is a very important part of the manuscript resuming the mechanism proposed by the authors and it should be mention along the discussion (maybe as a conclusion?)
Accepted. Some parts of the figure 4d legend (e.g. MOMP) are already mentioned in the Discussion. Now we added also a conclusive sentence on a possibly same mechanism of CsA and TAM on CypD, which prevent the integration of the latter into the mPTP (lines 279-280) and the impact of a sustained mitochondrial Ca2+ overload on mitochondrial metabolism (lines 295-297). We consider Fig. 4, which copies the graphical abstract, per se as a form of conclusion and prefer to keep it this way, without a duplication of conclusions in body text.
Methods
- Sections 4.5 and 4.9 should be fused, since they described the transfection of different constructs and seems repetitive.
Agree, done as suggested

Reviewer 2 Report
T- ALL remains an highly aggressive disease with poor prognosis and high relapse rate, due to development of multiple mechanism of resistance to conventional and often to more innovative chemotherapy.
To understand resistance pathways, and discover potential cellular targets able to counteract it, can open the way to new drug combinations aiming to improve the poor prognosis of this disease. Unfortunately, the “de novo” anticancer drug development is an expensive and time- consuming process, that, despite huge investments and work, often end in failure, explaining the low number approved drugs each year by regulatory agencies.
On this basis drug repurposing, repositioning or reprofiling represent a very promising strategy to speed up the new therapeutic options for cancer patients. Besides the obvious financial advantage, repurposing drugs often avoid the need for additional studies to investigate pharmacokinetic properties and toxicity.
I this paper the authors proponed the combined use of two, old, FDA-approved drugs, currently used in the neurological disorder treatment ( cannabidiol, CBD) and in ER+ breast cancer tamoxifen, TAM), as antileukemic drugs in T-ALL. They performed a series of in vitro experiments using two T-ALL derived cell lines, with the purpose to elucidate the mechanisms of their synergistic effects and to identify the ideal administration sequence to maximize the cytotoxic effect. They demonstrated that CBD and TAM affects mitochondria, even with different molecular target depending on the sequence of drug application, and the models illustrated in figure 4 is very appealing and clear.
Even more interesting is the identification of TAM and TAM derivative binding sites on CypD, which potentially pave the way to the development of new small molecules able to interact with CypD.
In general, all the paper is clear, experiments well conducted, described and illustrated. Discussion is exhaustive and references updated.
I would be useful to know the activity of the proposed drug combination in the normal T-cell counterpart, and in T- all cells. Have the authors any idea on this? How about toxicity on normal tissues/cells?
Author Response
T- ALL remains an highly aggressive disease with poor prognosis and high relapse rate, due to development of multiple mechanism of resistance to conventional and often to more innovative chemotherapy.
To understand resistance pathways, and discover potential cellular targets able to counteract it, can open the way to new drug combinations aiming to improve the poor prognosis of this disease. Unfortunately, the “de novo” anticancer drug development is an expensive and time- consuming process, that, despite huge investments and work, often end in failure, explaining the low number approved drugs each year by regulatory agencies.
On this basis drug repurposing, repositioning or reprofiling represent a very promising strategy to speed up the new therapeutic options for cancer patients. Besides the obvious financial advantage, repurposing drugs often avoid the need for additional studies to investigate pharmacokinetic properties and toxicity.
I this paper the authors proponed the combined use of two, old, FDA-approved drugs, currently used in the neurological disorder treatment ( cannabidiol, CBD) and in ER+ breast cancer tamoxifen, TAM), as antileukemic drugs in T-ALL. They performed a series of in vitro experiments using two T-ALL derived cell lines, with the purpose to elucidate the mechanisms of their synergistic effects and to identify the ideal administration sequence to maximize the cytotoxic effect. They demonstrated that CBD and TAM affects mitochondria, even with different molecular target depending on the sequence of drug application, and the models illustrated in figure 4 is very appealing and clear.
Even more interesting is the identification of TAM and TAM derivative binding sites on CypD, which potentially pave the way to the development of new small molecules able to interact with CypD.
In general, all the paper is clear, experiments well conducted, described and illustrated. Discussion is exhaustive and references updated.
I would be useful to know the activity of the proposed drug combination in the normal T-cell counterpart, and in T- all cells. Have the authors any idea on this? How about toxicity on normal tissues/cells?
We gratefully acknowledge the Reviewer’s opinion. When it comes to the question, raised at the end, we can provide the following information. 1. In our previous papers (Olivas-Aguirre et al., 2019; Torres-Lopez et al., 2019) we have demonstrated a preferential suppression of cancer cells, especially, T-ALL, by CBD and TAM. However, CBD, which shows insignificant effect on resting healthy T-cells, affects mitogen-activated T-cells similarly to T-ALL, thus it appears to work harder on cells with increased metabolism. 2. Currently, we have no data on the effect of this drug combination on healthy tissues or T-lymphocytes. We will be working on it. The present study focuses on the molecular mechanism of TAM/CBD combined effect. We have not included in this instance a large body of preliminary data on the triple combination of DEX/TAM/CBD as well as data on the impact of TAM/CBD induced mitochondrial alterations on ROS production and autophagy. The current granted project envisages intensive preclinical studies and we hope to succeed in this phase, offering an argumented therapy approach, which, as a part, will address safety issues/side effects.